# ShiftMorph: A Fast and Robust Convolutional Neural Network for 3D Deformable Medical Image Registration

## ABSTRACT

Deformable image registration (DIR) is crucial for many medical image applications. In recent years, learning-based methods utilizing the convolutional neural network (CNN) or the Transformer have demonstrated their superiority in image registration, dominating a new era for DIR. However, very few of these methods can satisfy the demands of real-time applications due to the high spatial resolution of 3D volumes and the high complexity of 3D operators. To tackle this, we propose losslessly downsampling by shifting the strided convolution. A grouping strategy is then used to reduce redundant computations and support self-consistency learning. As an inherent regularizer of the network design, self-consistency learning improves the deformation quality and enables halving the proposed network after training. Furthermore, the proposed shifted connection converts the decoding operations into a lower-dimensional space, significantly reducing decoding overhead. Extensive experimental results on medical image registration demonstrate that our method is competitive with state-of-the-art methods in terms of registration performance, and additionally, it achieves over 3× the speed of most of them.

## CCS CONCEPTS

• **Computing methodologies** → **Computer vision problems**.

## KEYWORDS

Deformable image registration, Fast, Robust, Shift, Self-consistency

## 1 INTRODUCTION

Image registration is commonly used in various biomedical applications, such as surgical guidance, histological imaging, and neurosurgery. This technique is often used as a preliminary step, particularly in applications that involve multiple misplaced images. For instance, registration is highly valued in surgical navigation systems for its potential to reproduce high-quality intraoperative images [4, 15, 26]. Image registration usually involves two stages, i.e., affine image registration (AIR) and deformable image registration (DIR). AIR aligns view perspectives through scaling, translation, and rotation in rigid scenarios [7, 11, 34, 43]. Nevertheless, DIR has to be conducted to align anatomical structures accurately because of continuous deformations of soft tissues and organs within the human body.

*ACM MM, 2024, Melbourne, Australia*
© 2024 Copyright held by the owner/author(s). Publication rights licensed to ACM.
ACM ISBN 978-x-xxxx-xxxx-x/YY/MM
https://doi.org/10.1145/nnnnnnn.nnnnnnn

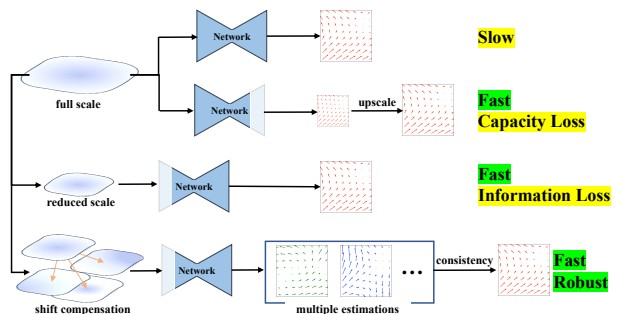

**Figure 1: Four pipelines of learning-based DIR. The proposed (bottom) one has both robust and high-efficiency advantages.**

Mathematically, DIR can be reduced to optimizing the deformation field that specifies a warping destination for each voxel. Previous studies have devised numerous non-rigid deformation models grounded in diverse theoretical frameworks. Thereinto, traditional machine learning methods [3, 41, 47, 49] iteratively optimize deformation fields based on hand-crafted energy functions. These techniques typically require numerous iterations to achieve desirable registration outcomes, resulting in low computational efficiencies.

In recent years, the boom of deep learning has opened up new avenues for image registration. Many learning-based techniques [2, 9, 11, 20, 33, 50] have now been introduced to enhance registration quality. However, most existing methods still can not fulfill the requirements of real-time scenarios, especially when handling 3D medical images. Despite GPU support, the inference time often remains impractical for time-critical applications, primarily due to the high resolution of 3D volumes and the high computational complexity of 3D operators. Seriously, the computational overhead explodes significantly as the resolution scales up. The computational modules at the top resolution level occupy the most inference time. Therefore, downsampling has to be used to reduce the image scale for some high-complexity models [29, 44, 53]. Some works [22, 46, 49] instead restrict the representation space of deformation fields. However, they cost the price of reduced image information or model capacity, as shown in Fig. 1.

Naively downsampling can cause a notable degradation in registration performance. The non-invertible downsampling process inevitably loses partial image details. To tackle this, we introduce the particular property of the strided convolution, i.e., the outcome sequence varies with the convolutional starting point. Hence, shifting the input sequence leads to distinct encoding results, thus complementing the missing information caused by scaling down. We then propose grouping the features according to their shifting behaviors, offering a basis for self-consistency learning. When merged within groups, the shifted encoding features can effectively

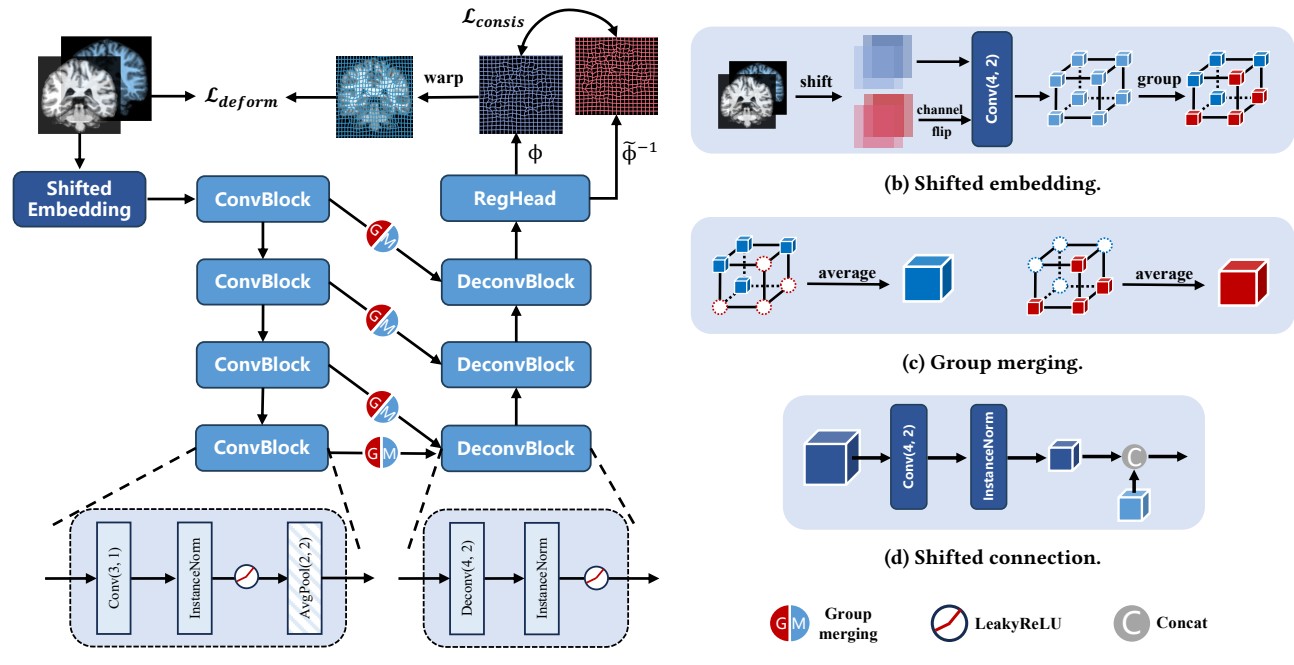

Figure 2: The overall architecture of ShiftMorph and the companion modules. The parameters within brackets indicate the kernel size and stride. The shifted embedding outputs eight feature tensors for each volume pair. These features are encoded through four convolutional blocks whose outputs are then merged within groups and concatenated to lower-level deconvolutional blocks. Average pooling is used only in the first three convolutional blocks. The registration head produces two deformation fields for the corresponding groups through two consecutive convolutional blocks and one plain convolution. Note that the input volume pair of the second group is reversed, resulting in an inverse registration direction. The two outcomes of the registration head are ultimately scaled up to match the original resolution.

remove redundant computations and improve robustness. Eventually, the grouping strategy and self-consistency learning allow for the trained network to be pruned by half after training without compromising the registration performance. Additionally, shifted connections restrict the decoding process in lower-dimensional spatial space, further accelerating the proposed network.

To summarize, the network proposed in this study is referred to as ShiftMorph, and the major contributions of this paper are as follows:

- We propose a lossless embedding module to reduce the spatial resolution and preserve image information simultaneously. We then present the companion modules, including feature grouping, group merging, and shifted connection, for fast and robust 3D image registration.
- Self-consistency learning, derived from our model design, assists in training the network and restricting the deformation difference between groups. The network can then be cut in half by discarding one group after training, thus doubling the throughputs of the proposed network.
- Extensive experimental results demonstrate the superiority of the proposed method in both runtime and registration performance. Our method achieves a significant speedup

over state-of-the-art methods without compromising the registration quality.

## 2 RELATED WORK

### 2.1 Deformable Registration

DIR is a voxel-level (or pixel-level for a 2D scenario) task that estimates a new coordinate for each voxel. Afterward, the moving image is warped and registered by interpolating under the obtained coordinate grid, which can be expressed as $I_w = \phi \circ I_m$. Here, $I_m$, $I_w$, and $\phi$ represent the moving image, the warping result, and the deformation field, respectively. Commonly, deformation refers to a natural process of continuity, differentiability, invertibility, and strong local consistency. As a result, the objective function for solving a registration problem typically consists of two terms:

$$\min_{\phi} \mathcal{L}_{\text{sim}}(I_f, \phi \circ I_m) + \lambda \mathbf{Reg}(\phi), \qquad (1)$$

where $I_f$ is the fixed target image; $\mathcal{L}_{\text{sim}}(\cdot)$, as a similarity metric, penalizes the distance between the registered image pair; $\mathbf{Reg}(\cdot)$ imposes extra constraints based on a priori knowledge of the deformation. Due to the physical nature of deformation, many methods

[5, 10] employ the diffusion energy and bending energy to regularize the first and second derivatives of displacement fields, promoting smooth deformation.

## 2.2 Diffeomorphic Deformation

Image registration is often utilized to facilitate information fusion for multiple images [21, 45], highlighting the importance of preserving topological structures. Thus, DIR often necessitates a delicate balance between registration precision and diffeomorphism. Well-designed regularization can somewhat avoid destroying topological structures by voxel folding, but there is usually no guarantee of diffeomorphism. To tackle this, the Lie group has to be introduced to ensure diffeomorphic deformation. Let deformation be the continuous motion within a time interval, i.e., $t \in [0, 1]$. $\phi_t$ represents the deformation field at a given time $t$. Assume that the velocity field, denoted by $v$, is stationary, which holds $vdt = d\phi_t$. In particular, $\phi_{t=0} = \mathbf{id}$ is an identity map. Then, the exponential map of the velocity field is defined by:

$$\exp(vt) = \lim_{n \to \infty} (\mathbf{id} + \frac{vt}{n})^n, \qquad (2)$$

which splits the deformation into $n$ equivalent minimal motions over $\frac{t}{n}$. Thus, $\phi_t$ belongs to a Lie group that holds $\phi_t = \exp(vt)$ and $\phi_{t_1+t_2} = \exp(vt_1) \circ \exp(vt_2)$. As long as each sub-deformation is sufficiently small, the integrated deformation can be ensured to be diffeomorphic. Ultimately, $\phi_1$ can be derived recursively through $\phi_t = \phi_{t/2} \circ \phi_{t/2}$, which is the widely applied squaring and scaling (SS) skill [1, 9, 32].

## 2.3 Learning-based Deformable Registration

Learning-based methods have been driving the advancement of DIR for years due to their capability of generalizing knowledge from vast amounts of data. These methods can be trained efficiently in end-to-end manners with appropriate loss functions. Currently, there are two main categories of registration networks according to their computational modules, i.e., CNN-based methods and Transformer-based methods. Both of the two types of registration networks commonly adopt UNet-like architectures [35] to produce deformation fields by incorporating a final registration head [2, 6, 37, 46]. Some studies [20, 28, 29, 39] propose refining the deformation field progressively, which is beneficial for addressing large deformations. Jia et al. [22] utilize the band-limited Fourier domain to represent displacement fields in a low-dimensional space. Kim et al. [24] propose employing cycle consistency to enhance topology preservation. Zhou et al. [52] and Vray et al. [42] utilize the technique of knowledge distillation to improve network efficiency and registration performance. The networks proposed by Chen et al. [5, 6, 38, 48, 53] introduce the widely applied self-attention mechanism to enlarge reception fields, achieving state-of-the-art registration performance. TransMatch [8] performs feature matching between the misplaced image pair with a modified cross-attention mechanism. Most of the Transformer-based methods adopt the attention technique of Swin-Transformer [27] due to the high resolution of 3D medical volumes.

## 3 METHODOLOGY

Although the existing methods have significantly improved registration performance, their computational complexities remain infeasible to deploy in real-time medical devices. Based on our observation that most computational latencies stem from the upper encoders and decoders, the main idea of this study is to sink the computation of the network down to a lower resolution while still maintaining high registration performance. The overall architecture and the proposed modules are illustrated in Fig. 2.

## 3.1 Shifted Embedding

A 1D sequence can be cut in half through the 1D convolution with a stride of 2, commonly used as a learnable downsampling approach. However, this process inevitably incurs image degradation and information loss, which is not invertible, making it impossible to recover the original information completely. Essentially, the strided convolution skips parts of the sliding positions, thus obtaining a shorter scanning path. The missing sliding positions can be directly recovered through shifting. As illustrated in Fig. 3, two different paths are derived by shifting the starting point, leading to distinct kernel coverages and convolutional outputs. Hence, the 1D stride-2 convolution can produce two different downsampling sequences, effectively filling in the missing details caused by resolution reduction. When extended to the 3D convolution, eight shifting options are available on the cubic grid.

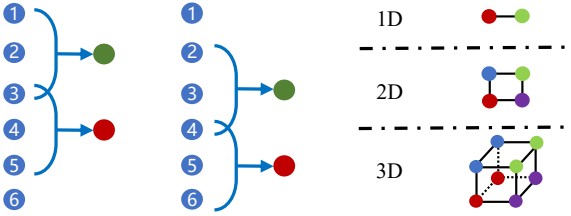

Figure 3: Two different scanning paths of the 1D stride-2 convolution. The left path is $\{1, 3\}$, whereas the right one is $\{2, 4\}$. The number of shifting options is 4 for the 2D convolution and 8 for a 3D scenario.

Based on this, a lossless downsampling operator is obtained by shifting the input images and acquiring multiple encoding sequences through the stride-2 convolution. We utilize this approach to construct the shifted embedding, which serves as the kernel module of the proposed network. As shown in Fig. 2b, the input image pair is concatenated and shifted along three directions to produce eight data volumes. The stride-2 convolution is then used to reduce the resolution to $\frac{1}{2}$. Nonetheless, this module actually performs pseudo-downsampling that grafts the spatial dimension onto the batch size. The complexity of network computation remains unchanged. Thus, we propose the complementary techniques of feature grouping and group merging to achieve efficient registration.

## 3.2 Feature Grouping and Group Merging

As commonly known, the local intensities of a natural image are highly correlated, which limits the encoding differences of the eight

shifts. In the decoding stage, i.e., the right part of Fig. 2a, which is comparatively costlier than encoding, reserving all these feature tensors is unnecessary. A better deal is to perform mean estimation as it can markedly reduce computational overhead and boost the feature representations.

Therefore, we divide the eight feature tensors into two groups according to their convolutional starting points. Fig. 2b displays that every four features with the same color are grouped together. Precisely, four shifting destinations of one group are positioned near the starting point, whereas others are far from it. The shifting step is minor, no greater than 1. Consequently, the two group-wise mean estimates exhibit a high level of consistency, thereby providing a basis for conducting self-consistency learning. In addition, averaging significantly reduces feature variance, demonstrating strong robustness.

During decoding, we initially perform averaging within groups to reduce the eight feature tensors to only two, thereby reducing the decoding overhead to $\frac{1}{4}$. Even though the encoding stream costs the same as before, the proposed shifting and grouping strategies significantly accelerate our network, in view of the fact that decoders of a UNet-like network are usually more expensive than encoders.

### 3.3 Shifted Connection

The typical implementation of the decoder with a skip connection can be formulated by:

$$f_{\text{dec}}^{2r} = W_{\text{dec}}^T(f_{\text{enc}}^r \oplus f_{\text{dec}}^r), \qquad (3)$$

where $\oplus$ and $W_{\text{dec}}^T$ represent concatenating and decoding, respectively; $f_{\text{enc}}^r$ and $f_{\text{dec}}^r$ denote the encoding and decoding outcomes at the resolution scale of $r$. The skip connection concatenates the decoding features with the encoding features at the same resolution level, which is time-consuming due to the concurrent expansion of channels and spatial resolution. We propose shifting the skip connection down to obtain an even cheaper decoder, i.e., the shifted connection in Fig. 2d. The shifted connection densely connects the decoding features with the downsampled features from the upper encoder, as expressed by:

$$f_{\text{dec}}^{2r} = W_{\text{dec}}^T(W_{\text{down}}f_{\text{enc}}^{2r} \oplus f_{\text{dec}}^r), \qquad (4)$$

where $W_{\text{down}}$ represents downsampling. As a result, feature concatenations are ahead of upsampling; the decoder operates in a lower-dimensional space, further reducing the computational complexity. Besides, this design is of lower rank than the classical skip connection, which is beneficial for filtering out noise and redundant features [14, 23].

### 3.4 Self-Consistency

Due to the grouping strategy, the final registration head of Shift-Morph can produce two deformation fields for each image pair, as depicted in Fig. 2a. These two fields, denoted by $\phi$ and $\tilde{\phi}$, represent the deformation of an approximately identical voxel grid. In view of this, a reasonable assumption is made that $\phi$ and $\tilde{\phi}$ are highly consistent. We then reverse the registration direction of the second group, thus obtaining $\tilde{\phi}^{-1}$. To promote natural deformations, an extra regularizer is introduced to approximate the identity map by

composing $\phi$ and $\tilde{\phi}^{-1}$, which is expressed by:

$$\mathcal{L}_{\text{consis}} = \|\phi \circ \tilde{\phi}^{-1} - \mathbf{id}\|_2^2. \qquad (5)$$

This regularization term, namely self-consistency learning, aids in model training and enhances the quality of deformation fields. In addition, the restricted difference between $\phi$ and $\tilde{\phi}$ offers a potential for removing one feature group after training.

### 3.5 Halved Network

In the training stage, both groups are needed for computing $\phi$ and $\tilde{\phi}$. Alternatively, either $\phi$ or $\tilde{\phi}$ can be used for testing due to self-consistency learning, which implies that we can discard one group naively after training. This paper only reserves the group near the starting point and uses $\phi$ to perform registration in testing cases. Hence, the overall computational burden of ShiftMorph can be reduced by half, achieving a 2× speedup on top of the complete implementation of ShiftMorph.

### 3.6 Recurrent Registration

A plain CNN tends to be weak in capturing long-distance correlations. Without special designs, these networks are prone to failure in the case of large deformation. This topic is beyond the focus of this paper. As an alternative, we start with $\phi^{(0)} = \mathbf{id}$ and recurrently warp the moving image multiple times as follows:

$$\phi^{(n)} = \mathbf{ShiftMorph}_\theta(I_f, \phi^{(n-1)} \circ I_m) \circ \phi^{(n-1)}. \qquad (6)$$

$\theta$ denotes the learnable parameters of ShiftMorph.

### 3.7 Loss Functions

As presented in Eq. (1), the deformation loss used in this paper consists of two terms based on the image similarity and the smoothness of the deformation field.

**Image Similarity Metric.** Mean square error (MSE) is a common metric for quantifying the similarity between the model output and the ground truth in generative tasks, such as denoising and super-resolution [30, 40, 51]. However, MSE is sensitive to the voxel intensity. The image pairs in registration tasks often hold weak voxel-wise consistency, thus limiting the effectiveness of MSE in training registration networks. Instead, normalized cross-correlation (NCC) is not sensitive to the voxel intensity and is preferable for measuring texture and structure similarities. NCC can be formulated as:

$$\mathbf{NCC}(x, y) = \frac{\mathbf{Cov}(x, y)\mathbf{Cov}(x, y)}{\mathbf{Var}(x)\mathbf{Var}(y)}, \qquad (7)$$

where $\mathbf{Cov}(\cdot)$ and $\mathbf{Var}(\cdot)$ respectively estimate the covariance and variance of the input sequence. Local NCC (LNCC), which estimates the covariance and variance within a local patch, is a more common choice when dealing with image data.

**Smoothness Regularization.** The smoothness of deformation is encouraged by its physical nature. Let $u$ represent the displacement field of a 3D image that holds $\phi = \mathbf{id} + u$ and $\nabla_i \phi = 1 + \nabla_i u, \forall i \in \{x, y, z\}$. $\nabla_x$, $\nabla_y$, and $\nabla_z$ are the Laplacian operators estimating the spatial gradients along three directions. We utilize the diffusion energy to penalize the spatial gradients of the output displacement

**Table 1: Comparison of runtime and registration performances on two Brain MRI registration benchmarks. CTPP and GTPP represent the registration time per image pair on the CPU and GPU, respectively. Rankings are superscripted on the right.**

| | Params (M) | MACs (G) | Runtime Performance CTPP (ms)↓ | GTPP (ms)↓ | OASIS Dice↑ | HD95↓ | Folds (%)↓ | IXI Dice↑ | HD95↓ | Folds (%)↓ |
|---|---|---|---|---|---|---|---|---|---|---|
| **TransMorph** | 46.6 | 723 | $4312^4$ | $189^7$ | $0.8212^3$ | $2.0752^3$ | $1.1298^8$ | $0.7432^6$ | $3.7359^6$ | $1.9497^8$ |
| **TransMatch** | 70.7 | 753 | $6736^8$ | $282^8$ | $0.8148^4$ | $2.1005^4$ | $0.9384^6$ | $0.7437^4$ | $3.7164^5$ | $1.7490^7$ |
| **VoxelMorph** | 0.3 | 514 | $2553^3$ | $134^5$ | $0.7962^8$ | $2.3949^9$ | $1.3027^9$ | $0.7204^9$ | $3.8964^8$ | $2.1760^9$ |
| **LapIRN** | 1.2 | 1067 | $5013^6$ | $158^6$ | $0.8119^6$ | $2.2052^7$ | $1.0850^7$ | $0.7408^7$ | $4.0488^9$ | $1.7140^6$ |
| **PCNet** | 4.4 | 1982 | $19275^9$ | $680^9$ | $0.8241^2$ | $2.0116^2$ | $0.0603^2$ | $0.7502^2$ | $3.5940^3$ | $0.3239^2$ |
| **FourierNet** | 4.1 | 170 | $1101^2$ | $52^3$ | $0.7947^9$ | $2.2523^8$ | $0.4557^3$ | $0.7405^8$ | $3.4667^2$ | $0.4496^3$ |
| **ShiftMorph** | 6.7 | 164 | $\mathbf{911}^1$ | $\mathbf{39}^1$ | $0.8116^7$ | $2.1124^6$ | $0.7786^5$ | $0.7434^5$ | $3.6464^4$ | $1.4993^5$ |
| **ShiftMorph-diff** | 6.7 | 164 | $5396^7$ | $40^2$ | $0.8122^5$ | $2.1006^5$ | $\mathbf{0.0231}^1$ | $0.7476^3$ | $\mathbf{3.4131}^1$ | $\mathbf{0.0480}^1$ |
| **ShiftMorph ×3** | 6.7 | 493 | $4634^5$ | $127^4$ | $\mathbf{0.8266}^1$ | $\mathbf{1.9771}^1$ | $0.4736^4$ | $\mathbf{0.7510}^1$ | $3.7502^7$ | $1.0764^4$ |

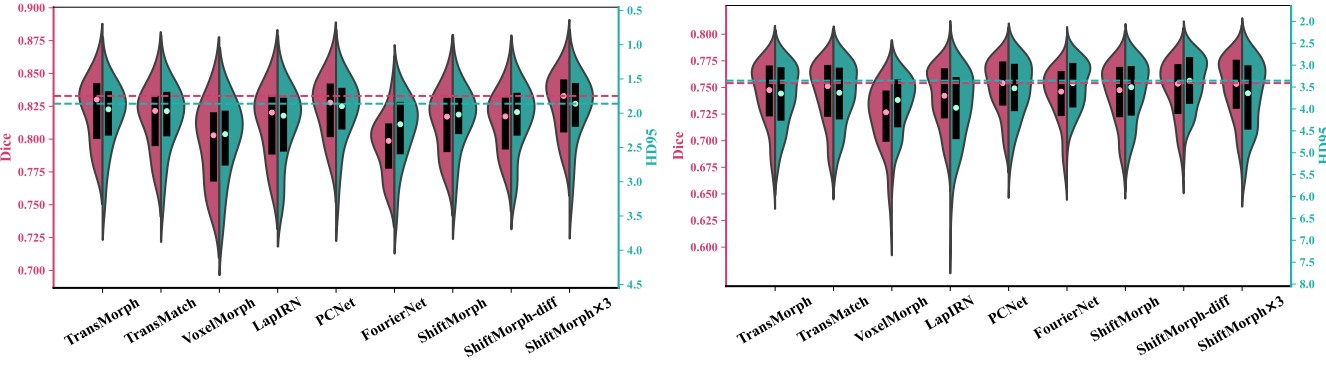

(a) Results on the OASIS dataset.          (b) Results on the IXI dataset.

**Figure 4: Violin plots with twin axes illustrating score distributions evaluated on two brain image registration tasks. Note that the HD95 axis is inverted for better visualization. ShiftMorph exhibits good registration performance and generalization ability, as proven by the highly clustered Dice and HD95 scores and consistent distribution shapes.**

field, as formulated by:

$$\mathcal{L}_{\text{Grad}}(\phi) = \frac{1}{3}\left(\|\nabla_x\phi - 1\|_2^2 + \|\nabla_y\phi - 1\|_2^2 + \|\nabla_z\phi - 1\|_2^2\right), \quad (8)$$

**Keypoint Correspondence Supervision**. Keypoint correspondences can be used as the ground truth to supervise deformation estimation. In this case, target registration error (TRE) is employed for supervised training. Consider two point clouds with $k$ matched point pairs of the fixed and moving image, denoted by $P_f = \{x_1, x_2, \cdots, x_k\}$ and $P_m = \{y_1, y_2, \cdots, y_k\}$. The TRE loss is then calculated as:

$$\mathcal{L}_{\text{TRE}} = \frac{1}{k}\sum_i^k \|x_i - y_i\|_2 \quad (9)$$

Incorporating the aforementioned self-consistency, the combined loss function of this paper is as follows:

$$\mathcal{L} = -\mathcal{L}_{\text{LNCC}}(I_f, \phi \circ I_m) + \alpha\mathcal{L}_{\text{TRE}}(P_f, \phi \circ P_m)$$
$$+ \lambda\mathcal{L}_{\text{Grad}}(\phi) + \gamma\mathcal{L}_{\text{consis}}(\phi, \tilde{\phi}). \quad (10)$$

# 4 EXPERIMENTS

## 4.1 Experimental Settings

**Baseline Methods.** We evaluate two Transformer-based methods and five CNN-based methods, including TransMorph [5], Trans-Match [8], VoxelMorph [2], VoxelMorph++[16], LapIRN[33], PCNet [29], and FourierNet [22], in comparison with ShiftMorph. Thereinto, PCNet, TransMorph, and TransMatch are the current state-of-the-art learning-based methods. VoxelMorph is a classical method and has been widely applied. VoxelMorph++ with spatial search for keypoint displacements is an improved version for supervised learning with keypoint correspondences. The original implementation of PCNet is based on TensorFlow. We reimplement it under the PyTorch framework to ensure a fair comparison. All the other methods are trained using the published source codes.

**Implementation Details.** Three versions of the proposed method are implemented in this paper, including ShiftMorph, ShiftMorph-diff, and ShiftMorph×3. ShiftMorph-diff employs the squaring and scaling skill to improve the diffeomorphism of the resultant deformation field through 7 iterations. ShiftMorph×3 recurrently warp the moving image three times and output $\phi^{(3)}$ following Eq. (6).

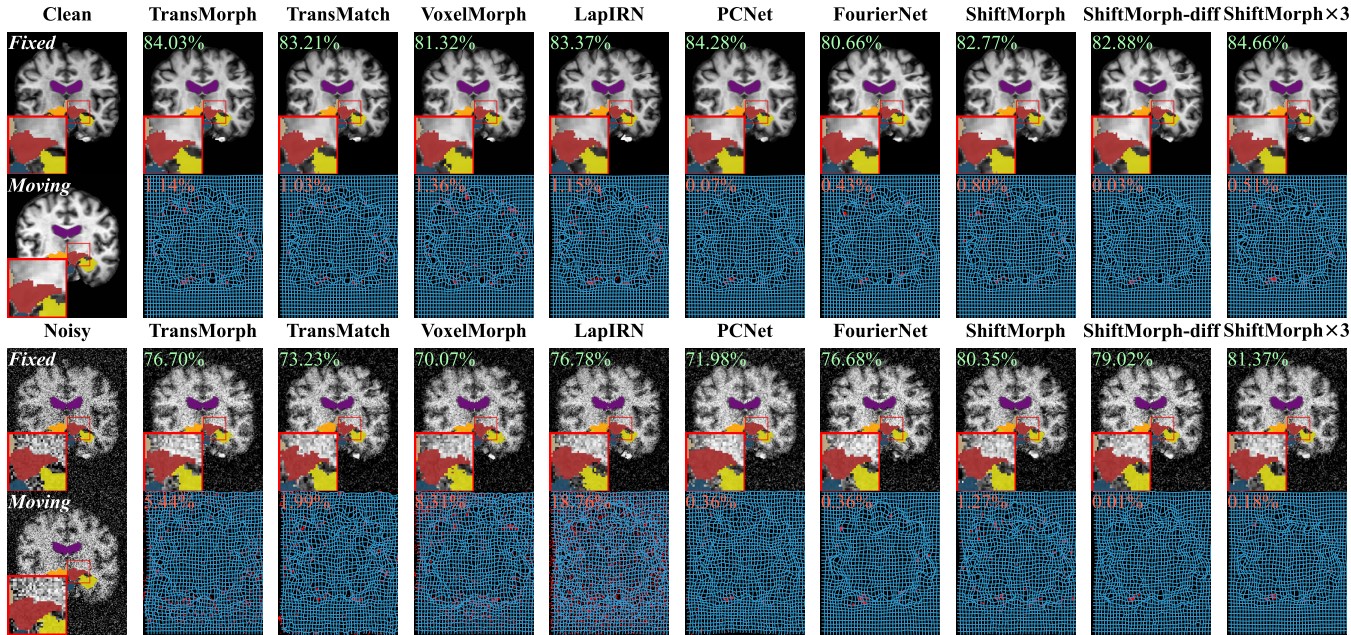

**Figure 5: Qualitative comparison of the warped brain slices and the deformation fields in normal and noisy (SD = 30) cases. The Dice scores are marked in green. The folded voxels and the fraction are marked in red. The registration results of our method are in the best accord with the target image and exhibit minimal noise effect, maintaining good shape continuity in the warping results and high smoothness in the deformation fields.**

**Parameter Settings.** The proposed method and the comparison baselines are trained under the same experimental settings for a fair comparison. We set $\alpha = 1$ for supervised training if correspondence labels are available; otherwise, TRE loss is not involved. We follow a common setting of $\lambda = 1$ without hyperparameter fine-tuning for all the comparing methods. $\gamma = 0.01$ is used only for ShiftMorph with self-consistency learning. The Adam optimizer [25] with a learning rate of 0.0001 is utilized for training. The learning rate is adjusted by a cosine annealing schedule with a maximum epoch of 300. All the experiments are conducted using an Intel Xeon Silver 4314 CPU and a 24G NVIDIA Geforce RTX 3090 graphics card.

**Evaluation Metric.** The Dice similarity coefficient (DSC) and Hausdorff distance (HD) are utilized to measure the registration accuracy of each anatomical structure. DSC calculates the degree of overlap of segmentation labels after deformation. The HD score measures the shape distance between the warped image and the target image from being isometric. We use the 95th percentile HD to mask out outliers, namely HD95. Topological preservation should also be emphasized to protect the original information of the moving image. To this end, the Jacobian Determinant (JacDet) is utilized to examine whether a voxel is folded by warping. $\nabla_x\phi(p)$, $\nabla_y\phi(p)$, and $\nabla_z\phi(p)$ represent 3 directional vectors determining the $p$-th minimal cube in $\phi$. The determinant, i.e., $|\nabla_x\phi(p), \nabla_y\phi(p), \nabla_z\phi(p)|$, calculates the volume of this cube. A non-positive volume value indicates that the corresponding voxel is flipped over or folded after deforming, losing its original topological structure. Thus, the degree of topology destruction is measured using the percentage of voxels with a non-positive JacDet.

## 4.2 Unsupervised Brain MRI Image Registration

In this section, we evaluate the registration performance in inter-patient and atlas-based brain magnetic resonance imaging (MRI) registration tasks.

For inter-patient registration, we utilize the OASIS dataset [19, 31] acquired from the 2021 Learn2Reg challenge [18], which consists of 451 brain T2-weighted MRI images. The Learn2Reg official split contains 394, 19, and 38 images for training, validation, and testing. The segmentation labels of testing images are not available. Thus, we conduct the testing stage on the validation set instead.

For atlas-based registration, we choose the IXI dataset[1], a commonly chosen benchmark. In this experiment, the brain atlas image is used as the source image, whereas patient images act as the target images. The IXI dataset consists of 414 brain MRI images that are T1-weighted. The original MRI scans are preprocessed using FreeSurfer [13] through a stream that includes skull stripping, spatial normalization, labeling, and others. We adopt the same split used by TransMorph [5], which contains 403 images for training, 58 for validation, 115 for testing, and one atlas image. Each image undergoes voxel-wise labeling for 30 anatomical structures.

The first part of Tab. 1 displays the runtime performance of registering images sized $160 \times 192 \times 224$ from the two datasets. The Transformer-based methods own large amounts of parameters and multiply-accumulate (MAC) operations, whereas CNN-based methods tend to be faster and contain fewer parameters. Most of the comparing methods incur high latencies exceeding 0.1s, leading to

---

[1]https://brain-development.org/ixi-dataset/

**Table 2: Comparision of runtime and registration performance for lung CT images sized** $160 \times 128 \times 160$ **in the Lung250M-4B dataset. The left part presents the required training time, memory usage, and inference time per pair on the GPU. TRE opt. represents the TRE score after instance optimization.**

| | Runtime Performance | | | Validation | | | Testing | | |
|---|---|---|---|---|---|---|---|---|---|
| | Tr. Time (h) | Tr. Mem. (GB) | GTPP (ms) | TRE (mm) | TRE opt. (mm) | Folds (%) | TRE (mm) | TRE opt. (mm) | Folds (%) |
| **TransMorph** | 7.50 | 8.25 | 86[7] | 5.1165[7] | 2.5871[8] | 1.1951[8] | 5.6740[7] | 2.3389[7] | 0.5618[8] |
| **TransMatch** | 7.58 | 9.25 | 103[8] | 6.4467[9] | 3.2218[9] | 0.5532[7] | 7.6717[9] | 4.2152[9] | 0.2711[7] |
| **VoxelMorph++** | 5.92 | 7.91 | 71[5] | 4.1250[3] | 2.2258[3] | − | 4.4215[3] | 1.9081[3] | − |
| **LapIRN** | 8.17 | 10.31 | 72[6] | 2.5124[2] | 2.0912[2] | 0.0230[3] | 1.9843[2] | 1.3338[2] | 0.0198[4] |
| **PCNet** | 12.25 | 11.76 | 320[9] | 5.1318[8] | 2.5779[7] | 0.1021[5] | 5.9177[8] | 2.6675[8] | 0.0565[6] |
| **FourierNet** | 5.92 | 4.34 | 25[3] | 4.6501[6] | 2.4452[6] | 0.1668[6] | 5.1303[6] | 2.2674[6] | 0.0546[5] |
| **ShiftMorph** | 5.58 | 6.24 | **19**[1] | 4.5318[5] | 2.3596[4] | 0.0522[4] | 4.9215[5] | 1.9989[5] | 0.0006[2] |
| **ShiftMorph-diff** | 5.67 | 6.72 | 23[2] | 4.4651[4] | 2.3606[5] | **0.0000**[1] | 4.7382[4] | 1.9358[4] | **0.0000**[1] |
| **ShiftMorph×3** | 8.01 | 12.62 | 61[4] | **2.2389**[1] | **2.0616**[1] | 0.0197[2] | **1.6953**[1] | **1.2883**[1] | 0.0180[3] |

a low throughput even on a high-performance GPU device. The results given by FourierNet seem appealing, but this method roughly masks out most high-frequency features, causing significant accuracy degradation. In comparison, ShiftMorph demonstrates significant superiority in both the computational overhead and the inference speed, achieving over 3× the speeds of most comparing methods. Even if recurrently warping the moving image three times for high accuracy, ShiftMorph×3 can still achieve a better speed than most on a GPU device. However, warping as a highly parallel operator is slow when using a serial CPU processor.

The right parts of Tab. 1 present the averaged evaluation results on these two registration tasks. Fig. 4 displays violin plots showing the score distributions of evaluated methods. The Dice and HD95 scores of our method are on par with those of the state-of-the-art methods. Besides, ShiftMorph-diff with enhanced diffeomorphism can be even more competitive, outperforming most CNN-based methods except for PCNet. Moreover, the recurrent version of ShiftMorph×3 achieves the best Dice and HD95 scores in most cases and keeps a favorable inference speed.

As for topological preservation, ShiftMorph-diff is the most effective in this experiment. ShiftMorph folds 0.7786% of voxels on the OASIS dataset; this value can be lowered to 0.0231% by ShiftMorph-diff. As for the IXI dataset, the folded voxel proportion is reduced from 1.4993% to 0.048%. In contrast, PCNet, equivalently armed with the SS skill, folds more voxels than our method and has significantly higher computational complexity. FourierNet also shows good runtime performance and topological preservation, but its registration precision is relatively weak. If not using the SS and band-limit skills, i.e., excluding ShiftMorph-diff, PCNet, and FourierNet, ShiftMorph keeps better diffeomorphism than other methods, which benefits from self-consistency learning and merging the shifted features.

## 4.3 Supervised Lung CT Image Registration

We then use the Lung250M-4B dataset [12], a combined lung computed tomography (CT) image registration benchmark, to evaluate our method in a supervised large deformation scenario. This dataset contains 124 pairs of expiratory and inspiratory lung CT scans from different patients. The official split selects 97, 17, and 10 paired scans for training, validation, and testing. For the CT scans in the training set, keypoint correspondences are automatically generated by the

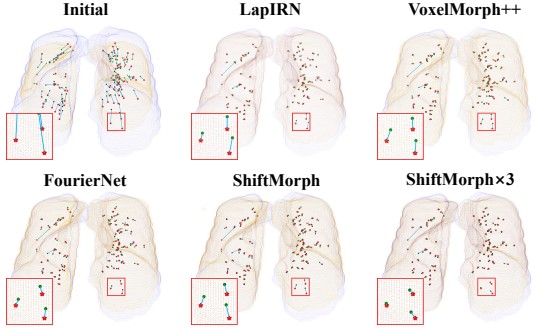

**Figure 6: Visualizing the deformation results of five representative methods without instance optimization. The yellow and blue masks represent the expiratory and inspiratory lung morphology, respectively. The proposed method performs the best in shortening the distances of matched landmarks in the region of lower lung lobes with larger deformations.**

corrfield method[2][36] to support supervised training. As for the validation and testing set, manual landmark annotations are provided to determine registration accuracy. Lung CT image registration is still challenging for learning-based methods. Following the pipeline in [36], we conduct instance optimization to refine the network outputs through 50 iterations of minimizing the dissimilarity of MIND features [17] and a Lapalace regularization.

Fig. 6 displays parts visualization results of representative methods. Numerical experimental results on this dataset are presented in Tab. 2. LapIRN and VoxelMorph++ show good registration accuracy in this case of large deformation due to the advantages of pyramid deformation enhancement and spatial search. Comparingly, ShiftMorph and ShiftMorph-diff, as plain CNN-based methods, get medium rankings. However, after recurrently composing the predicted deformation fields, ShiftMorph×3 can effectively produce long-distance deformations, achieving the best TRE scores on both the validation and testing stages. Moreover, ShiftMorph×3 exhibits appealing throughput and diffeomorphism, exceeding most comparing methods on this dataset.

---

[2]https://grand-challenge.org/algorithms/corrfield/

## 4.4 Robustness Testing

We next utilize Gaussian noise to simulate image degradation and conduct robustness testing on the OASIS dataset. Fig. 5 illustrates the warped images and deformation fields produced by corresponding methods. The baseline methods, except FourierNet, suffer from significant performance degradation, resulting in seriously folded deformation fields and abnormal warping results. In contrast, Shift-Morph demonstrates the best deformation results and preserves topological structures commendably, exhibiting smooth and continuous surface shapes in the warped slices. Fig. 7 exhibits the impacts of various noise levels on registration performances. The noise level indicates the standard deviations (SD) of Gaussian noise added to the testing images with a voxel intensity range of $[0, 255]$. As the noise level increases, the registration performance of most comparing methods decreases linearly with a large slope. In contrast, the Dice score of ShiftMorph only experiences a slight drop when the noise level is 10 and decreases with a small slope when the noise level exceeds 10.

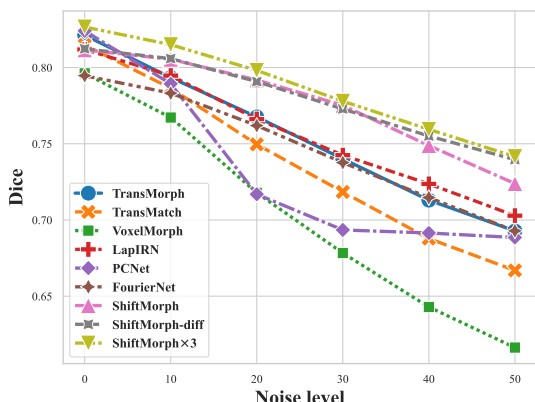

**Figure 7: The impacts of noise on registration performance regarding the Dice score. Our method exhibits good robustness to Gaussian noise.**

## 4.5 Ablation Study

We conduct a brief ablation study on the proposed modules and assess the effect of self-consistency learning. We evaluate the registration performance of the corresponding model variants on the OASIS dataset. Results are summarized in Tab. 3 and Tab. 4. By default, we apply the shifted embedding in combination with the grouping strategy to save space.

The first item of Tab. 3 represents a naive UNet using the skip connection without downsampling. Downsampling with trilinear interpolation significantly improves network speed; however, it comes at the high price of a loss of registration accuracy. Comparingly, shifted embedding armed with the grouping strategy can reduce computational overhead and improve registration performance. Besides, the voxel folding is mitigated by imposing consistency among neighbor voxels with self-consistency learning. The shifted connection also contributes to a notable improvement in network performance, achieving 1.6× the speed of a naive UNet.

Ultimately, ShiftMorph achieves 3.6× the speed, better registration results, and better diffeomorphism.

As evidenced by Tab. 4, performing self-consistency learning is beneficial for improving the quality of output deformation fields. In most cases, except the red one, self-consistency leads to better evaluation scores for three versions of ShiftMorph. However, this improvement is relatively small when combined with the SS skill, which can also be viewed as particular regularization. Applying too much regularization may cause underfitting.

**Table 3: Ablation study on different ways to downsampling and feature concatenation. × represents registration without downsampling; Tri. represents the naive downsampling using trilinear interpolation. "Skip" represents the classical skip connection. SEMB. and SCON. represent the proposed shifted embedding and shifted connection, respectively.**

| Down. | Connect | Dice | HD95 | Folds (%) | GTPP (ms) | Speedup |
|-------|---------|------|------|-----------|-----------|---------|
| × | **Skip** | $0.8084^4$ | $2.2367^4$ | $1.1630^4$ | 140 | - |
| × | **SCON.** | $0.8100^3$ | $2.1973^3$ | $1.2278^5$ | 88 | ×1.6 |
| **Tri.** | **Skip** | $0.7984^5$ | $2.2428^5$ | $0.9251^3$ | 19 | ×7.4 |
| **SEMB.** | **Skip** | $0.8109^2$ | $2.1729^2$ | $0.8922^2$ | 103 | ×1.4 |
| **SEMB.** | **SCON.** | $\mathbf{0.8116}^1$ | $\mathbf{2.1124}^1$ | $\mathbf{0.7786}^1$ | 39 | ×3.6 |

**Table 4: Ablation study on self-consistency for three versions of ShiftMorph. ✗ represents $\gamma = 0$; ✓ represents $\gamma = 0.01$. Improved scores are marked in blue.**

| | Self-Consistency | Dice | HD95 | Folds (%) |
|---|---|---|---|---|
| ShiftMorph | ✗ | 0.8102 | 2.1475 | 0.8738 |
| | ✓ | 0.8116 | 2.1124 | 0.7786 |
| ShiftMorph-diff | ✗ | 0.8120 | 2.0842 | 0.0256 |
| | ✓ | 0.8122 | 2.1006 | 0.0231 |
| ShiftMorph×3 | ✗ | 0.8258 | 1.9804 | 0.5052 |
| | ✓ | 0.8266 | 1.9771 | 0.4736 |

## 5 CONCLUSION

In this paper, we have investigated the common issue that 3D deformable registration networks frequently suffer from low inference efficiency. We have revisited the widely adopted network architecture of the UNet style for image registration, demonstrating that the major computational burden of these networks stems from the operations at the top level. To achieve fast and high-quality registration, we have proposed downsampling using the strided convolution and compensating for information loss by shifting. We have presented a grouping strategy to merge features group-wisely and reduce redundant computations. The proposed shifted connection contributes to a further acceleration and improvement of registration. Self-consistency learning, derived from the grouping strategy, promotes the quality of deformation fields and supports pruning half of the network. Experimental results demonstrate that the proposed method performs competitively with current state-of-the-art methods, however, at 3× the speed of most.

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
