# OpenReview forum: "ShiftMorph: A Fast and Robust Convolutional Neural Network for 3D Deformable Medical Image Registration"
_acmmm.org/ACMMM/2024/Conference — MM2024 Poster_

### Official Review · Reviewer_mhfS · 2024-05-03

**Rating:** 4
**Confidence:** 3

**Summary:**

This paper is about image registration using CNN. The main method is lossless downsampling by shifting the strided convolution. A grouping strategy is then used to reduce redundant computations by using the average of pixels in a local group to represent the region and support self-consistency learning. The process of taking the average of pixels in a local group has the advantage of reducing variance of features. Self-consistency learning is used to improve the quality of deformation. The final results are small network and fast decoding.

**Strengths:**

Main strengths include 1D stride-2 convolution to produce two downsampling sequences. Other strengths include the use of shifted connection as a way to reduce inference time. The overall idea of training the network to produce two similar deformation fields \phi and \tilde{\phi}^{-1} and \phi*\tilde{\phi}^{-1} are approximately the inverse of each other and then only one is needed for registration seems novel.

**Limitations:**

Limitations include some hard-to-understand sentences. For example, “they cost the price”, probably the authors meant to say “they pay the price” or “they cost”.

In Equation (8), it seems L_Grad is not normalized by the number of pixels in an image, but in Equation (9), L_TRE is normalized by the number of matched point pairs. Will it create a bias toward one loss or the other, depending on how many matched point pairs there are?
It seems some explanation about LNCC is needed. How is a local patch selected? How big is it?

Furthermore, how are the keypoints selected to compute TRE loss for the MRI? In other words, how are the point clouds computed?

How to determine the weight factors alpha, lambda, gamma in Equation (10)? The authors used alpha=1, lambda=1, and gamma = 0.01, are those choices applicable to other types of image?

It seems that \gamma (Table 4) has little effect on the performance of the proposed method. Then why is this loss term included in Equation (10)?

If not considering time of computation, will better results be obtained if one warp the moving image more than three times? That is, will people obtain better registration results by ShiftMorph*N, where N> 3?

**Suitability:**

2

---

### Official Review · Reviewer_3CCQ · 2024-05-28

**Rating:** 3
**Confidence:** 2

**Summary:**

This paper propose losslessly downsampling by shifting the strided convolution for 3D Deformable Medical Image Registration. Besides, a grouping strategy is then used to reduce redundant computations and support self-consistency learning. Experiments prove the effectiveness of the proposed method.

**Strengths:**

The article is well organized and elaborated, and the experiments are relatively rich.

**Limitations:**

1) Although the manuscript proposes a ShiftMorph network, it is not innovative enough compared with existing 3D deformable image registration methods;
2) There are many hyperparameters in Equation(10), making the parameter tuning process complex and time-consuming.

**Suitability:**

1

---

### Official Review · Reviewer_PUgJ · 2024-05-30

**Rating:** 3
**Confidence:** 4

**Summary:**

1. This paper proposes a lossless embedding module to reduce the spatial resolution and preserve image information simultaneously.
2. This paper proposes self-consistency learning to train the network and restrict the deformation difference between groups.
3. Experimental results show good performance in both runtime and registration performance.

**Strengths:**

1. The motivation of this paper is clear, it aims to give an efficient solution for deformable image registration.
2. The proposed methods for shifting are new, along with the merging and self-consistency learning design.
3. The comparison results are promising, and the experiments are relatively complete.

**Limitations:**

1. Relevance to multimedia is not clearly stated. I think the paper about medical image registration may be more suitable for MICCAI or ISBI conferences.
2. Some reference papers are missing. The proposed ShiftMorph relies on the novel operation of the input image, and I think the author should discuss the related paper "Cross-Resolution Distillation for Efficient 3D Medical Image Registration" in the Related Work section. Although numerical comparison is unnecessary due to different technical routes, a discussion is needed.
3. The specific network structure is not clear, including the number of conv blocks, channels and so on. By the way, why do we need to use instance normalization?
4. Visualization is not convincing. First, the differences in segmentation masks between different methods are not obvious. Second, the warped moving image seems to be distorted. Usually, such distortions are caused by unsuitable regularization, can you explain this problem?
5. The ablation experiments about regularization weights are missing. As mentioned above, the regularization weights are important, so the related experiments are demanded.
6. Though the performance compared with baseline methods is promising, the improvement of efficiency should be clarified in detail. In other words, I think the author should conduct one efficiency comparison experiment, focusing on the proposed operations and maintaining the same parts including conv blocks/channels as other baseline methods, aiming to show the superiority of the key contribution.
7. When applied with the diff-version, I wonder how to operate the self-consistency learning, can you give the specific designs?

**Suitability:**

2

---

### Meta-Review · Area_Chair_2Dbk · 2024-07-02

**Recommendation:** Accept (Poster)
**Confidence:** 4

**Metareview:**

Although two of three reviewers tend to accept this paper, and the rebuttal letter has addressed most concerns from the two reviewers. However, it does not address the concerns from the remaining reviewer, specially about the novelty.